# Coastal Upwelling in the Western Bay of Bengal: Role of Local and Remote Windstress

**Sthitapragya Ray** [1] **, Debadatta Swain** [1,*] **, Meer M. Ali** [2,3] **and Mark A. Bourassa** [2,4]

1   School of Earth, Ocean and Climate Sciences, Indian Institute of Technology Bhubaneswar, Argul, Jatni 752050, India
2   Center for Ocean-Atmospheric Prediction Studies, The Florida State University, Tallahassee, FL 32306, USA
3   Andhra Pradesh State Disaster Management Authority, Kunchanapalle 522501, India
4   Department of Earth, Ocean and Atmospheric Science, The Florida State University, Tallahassee, FL 32306, USA
*   Correspondence: dswain@iitbbs.ac.in

**Highlights:**

**What are the main findings?**

- High AWS and coincident strong negative UI$_{SST}$ (indicative of coastal upwelling) were observed along the western Bay of Bengal. The northern part of the coast illustrated UI$_{SST}$ leading ET compared to the south where both prevailed simultaneously.
- The equatorial windstress-forced first upwelling Kelvin wave triggered coastal upwelling along the northern part of the east coast of India in April, while the first downwelling Kelvin wave suppressed it in June.

**What is the implication of the main finding?**

- Coastal upwelling and hence, the seasonal variability of primary productivity in the region is not locally driven but influenced by remote equatorial windstress generated coastally trapped Kelvin waves.

**Abstract:** Monsoon winds drive upwelling along the eastern coast of India. This study examined the role of coastally trapped Kelvin waves in modulating the seasonal variability of local alongshore windstress (AWS)-driven coastal upwelling along the western Bay of Bengal. The winds generated AWS resulting in a positive cross-shore Ekman transport (ET) from March to the end of September, which forced coastal upwelling along the eastern coast of India. However, coastally trapped Kelvin waves could also modulate this process by raising or lowering the thermocline. Remotely sensed windstress, sea surface temperature (SST), and sea surface height anomaly (SSHA) were used to compute the AWS (the wind-based proxy upwelling index) and an SST-based proxy upwelling index (UI$_{SST}$). A new parametric method of the estimation of coastal angles was developed to estimate the AWS and ET. Coastal upwelling and the Kelvin waves were identified based on the climatology of SSHA, AWS, and UI$_{SST}$, in addition to a complex principal component (CEOF) analysis of the SSHA. The UI$_{SST}$ and AWS were found to be closely correlated along the southern section of the east coast of India (between Kavali and Point Calimere), where the coastal upwelling was largely local AWS-driven. However, along the northern section of the coast (between Kashinagara and Kakinada), coastal upwelling was triggered by the first upwelling Kelvin wave, sustained by the local AWS, and then terminated by the first downwelling Kelvin wave. This analysis illustrated that remote equatorial windstress caused coastal upwelling along the northern part of the Indian east coast, while it was primarily locally driven in the southern coast. The findings are helpful in better understanding the mechanisms modulating coastal upwelling along the western Bay of Bengal. These would provide useful insights into the primary productivity and the air–sea interactions in the region.

**Keywords:** coastal upwelling; upwelling index; scatterometer; wind stress; Kelvin wave; Ekman transport

## 1. Introduction

Coastal upwelling is associated with the upward flow of nutrient–rich subsurface waters into the euphotic layer as a result of the wind-induced divergence of surface currents along the coast. These upwelled waters, in turn, result in phytoplankton blooms which support large zooplankton populations along with larger marine organisms [1]. Coastal upwelling brings relatively cooler waters closer to the coast, hence the near-shore gradient of the sea surface temperature (SST) is often used as a proxy upwelling index (SST-based UI: $UI_{SST}$). Additionally, a positive alongshore windstress (AWS) (northward winds for east-facing coasts in the northern hemisphere) is also known to set up the cross-shore surface currents directed offshore (positive Ekman transport (ET)), thereby producing a coastal divergence of currents [2]. Subsequently, this divergence of the sea surface waters is replenished by waters from below the surface, which constitutes upwelling. Thus, the offshore ET could also serve as a proxy upwelling index based on windstress. In addition to this, negative coastal SSHA variations are known to be associated with coastal upwelling.

The upward flux of nutrient-rich waters associated with the upwelling process renders the major upwelling regions as some of the most productive parts of the global oceans. In fact, the major coastal upwelling systems associated with eastern boundary currents comprising only 1% of total ocean surface area contribute about 20% of global fish production [3]. In addition, understanding coastal upwelling is necessary, as it plays an important role in mediating various air–sea fluxes, including the air–sea $CO_2$ flux [4]. The major perennial coastal upwelling systems of the world, namely, California, Peru-Chile, Portugal–Northwest Africa, and Southwest Africa are called the Eastern Boundary Upwelling Systems (EBUS) [2]. Further, several seasonal coastal upwelling systems also exist along the eastern and western boundaries of various ocean basins.

The seasonal upwelling systems in the North Indian Ocean (NIO), like those along the coasts of Somalia, Oman, western and eastern India, Sri Lanka, and southeast Asia are associated with the seasonally reversing monsoon winds [2,5,6]. The seasonal upwelling system along the western margin of the Bay of Bengal (BoB) is driven by southwesterly monsoon winds which align parallel to the coast from March to April, with peak values during June to September, and which set up a narrow (30–40 km wide) upwelling zone along most of the coastline (11–18°N) [7]. Runoff from some of the major rivers such as the Ganga, Brahmaputra, and Mahanadi in the Indian subcontinent, which peaks during the later part of southwest (SW) monsoon, suppresses the upwelling signatures along the northern parts of the coast through barrier layer formation. The hydrographic analysis of [8] observed sea surface signatures were consistent with coastal upwelling at 18°N, 87°E (northeastern BoB) during September 1996. Similar observations of coastal upwelling were made by ref. [9] along the stretch of the east coast of India from 16°30′N to 18°30′N in hydrographic data. Coastal upwelling signatures have also been identified along the stretch of the coast between Chennai (13.082°N) and Vishakapatnam (17.687°N) based on satellite SST data [10]. Ref. [11] have analyzed the long-term variability of coastal upwelling along the eastern coast of south India using the $UI_{SST}$ and AWS; however, they have focused primarily on the local drivers. Ref. [12] has identified various forcings driving and modulating coastal upwelling in the NIO over different timescales. Mesoscale eddies and equatorial waves are found to be the significant drivers of coastal SSHA over intraseasonal and interannual timescales, respectively. However, over seasonal timescales, ref. [12] observed that coastal upwelling along the east coast of India is driven by both the local AWS and remote equatorial forcing.

The remote forcing in the BoB is comprised of a pair of upwelling and downwelling coastally trapped Kelvin waves that propagate counterclockwise from the coast of Sumatra,

northward along the eastern margin of the Bay. The first (second) upwelling Kelvin wave was propagated from January to April (August to September), followed by the first (second) downwelling Kelvin wave between April and August (October and December). These coastally trapped Kelvin waves also radiated westward propagating Rossby waves from the southeastern BoB into the interior of the Bay. In addition to these, Rossby waves were also generated in the interior BoB through windstress curl [13]. Refs. [12,14] have noted that the BoB circulation on seasonal timescales is determined by both local and remote wind forcing (following linear wave theory). We have examined the relative importance of these two forcing mechanisms in driving the western BoB (premonsoon and SW monsoon) coastal upwelling system in the current study. Additionally, the spatiotemporal variability of the coastal upwelling system along the eastern coast of India during the SW monsoon based on several remotely sensed reanalysis datasets was also analyzed.

## 2. Data

The analysis was carried out utilizing three sets of remote-sensing reanalysis datasets spanning the period from 2009 to 2018 (Table 1). These datasets consisted of 6-hourly winds from the Institut Français de Recherche pour l'Exploitation de la Mer (IFREMER), daily Operational SST and Ice Analysis (OSTIA) SST, and Copernicus Marine Environment Monitoring Service (CMEMS) gridded daily SSHA. The 6-hourly gridded winds at a grid spacing of 0.25° are available from IFREMER through CMEMS. The 10 m L2B equivalent neutral winds swath data from multiple scatterometer missions (ERS-1 to Scatsat-1) are combined with ancillary data from radiometers (SSM/I, SSMIS) and atmospheric wind reanalysis (ERA-interim or ERA-5) to generate this gridded gap-free wind product. We have used zonal and meridional wind stress data obtained from these datasets. The product user manual and quality information document for this dataset are available through CMEMS [15]. Intercomparison of these wind products with ERA-5 wind vectors yielded a wind speed root mean square difference (RMSD) of 1.87 m/s close to the coast, and the vector correlation between the two wind direction data was 1.63 near the coast (out of a maximum possible value of 2.0) [16].

**Table 1.** Details of datasets used.

| Parameter | Dataset | Grid Spacing | Temporal Resolution |
|---|---|---|---|
| Zonal and Meridional Windstress | CMEMS WIND_GLO_WIND_L4 _REP_ OBSERVATIONS_012_006 | 0.25° × 0.25° | 6 h |
| Sea Surface Temperature | OSTIA Near Real Time Level 4 SST | 0.05° × 0.05° | Daily |
| Sea Surface Height Anomaly | CMEMS SEALEVEL_GLO _PHY_L4_MY_008_047 | 0.25° × 0.25° | Daily |

The OSTIA L4 daily SST provides gridded SST data at a grid spacing of 0.05° generated from the combination of observations from multiple infrared and microwave sensors [17]. The product user manual [18] and quality information document [17] for this dataset are available through CMEMS. The product has a mean difference of 0.06 °C for the global oceans, and 0.02 °C for the Indian Ocean in comparison with in situ SST observations from ARGO floats [17].

SSHA datasets from SEALEVEL_GLO_PHY_L4_ MY_008_047 generated by SL-CLS-TOULOUSE-FR and distributed by CMEMS are used in this analysis. These daily data products are generated at 0.25° grids from multiple altimeters. At any given time, this dataset combines lower level or reprocessed observations from all available altimeters at that time from a list of about 40 altimeter missions following the methodology of [19]. Ref. [20] validated this product against along-track Saral/Altika SSHA observations and reported RMS differences ranging from 1 cm in low-variability areas to 18 cm in high-variability areas and coastal errors of 7 cm. Within the BoB, the RMS difference is fairly low

in the central and western BoB, while the greatest errors are observed along the eastern margin and the head of the Bay. The effective resolution of this product was estimated by ref. [20] following the spectral coherence approach and the values for the BoB range from 250 to 300 km.

## 3. Methodology

The analysis involved the proper identification of the spatial and temporal extent of the coastal upwelling processes along the Indian coast as the preliminary step [7–12]. Consequently, the coastline between 10.25°N and 19°N, as well as from 78°E to 90°E, with upwelling features was identified for carrying out the intended analysis. In addition to these coastal points, the entire Bay of Bengal (5–25°N, 78–100°E) was selected as the study area for the complex empirical orthogonal function (CEOF) analysis. Windstress data were sampled to a daily temporal resolution centered on the date of the SSHA dataset. For this, a mean of the high-temporal-resolution data (windstress) over the period of the low-resolution data (SSHA) was considered (after filtering high frequencies) whenever a direct comparison was necessary.

The coastline was first extracted from the GSHHG coastline dataset. Subsequently, the coastline was simplified by manually rejecting points where multiple points were present in close proximity to the complex coastal features of scales comparable to a 0.25° distance. In the case of the enclosed bays, the inner points were rejected while the outer ones were retained. This allowed us to obtain distinct coastal locations for the analysis, thus resulting in a set of 23 coastal points for the present study (Figure 1). The AWS, UI$_{SST}$, and SSHA were sampled at each of these coastal points and their decadal climatology (2009–2018) were analyzed, focusing on the premonsoon (March–May) and SW monsoon (June–September) seasons. In addition, the entire spatial SSHA dataset over the full 2009–2018 period (not restricted to premonsoon and monsoon seasons) was also used to carry out a CEOF analysis to investigate the role of coastally trapped Kelvin waves in modulating this upwelling system. These methods are further described in the following subsections. Figure 2 presents a schematic of the entire methodology used in the present analysis.

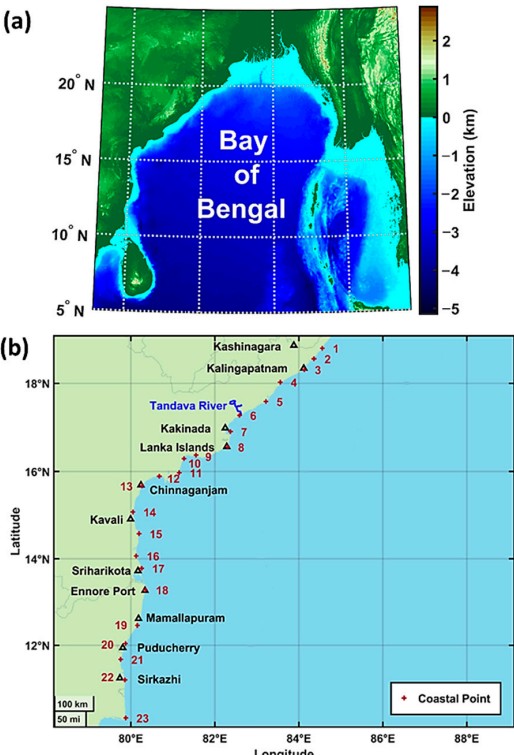

**Figure 1.** (**a**) Study area with overlaid bathymetry; (**b**) Coastal observation points.

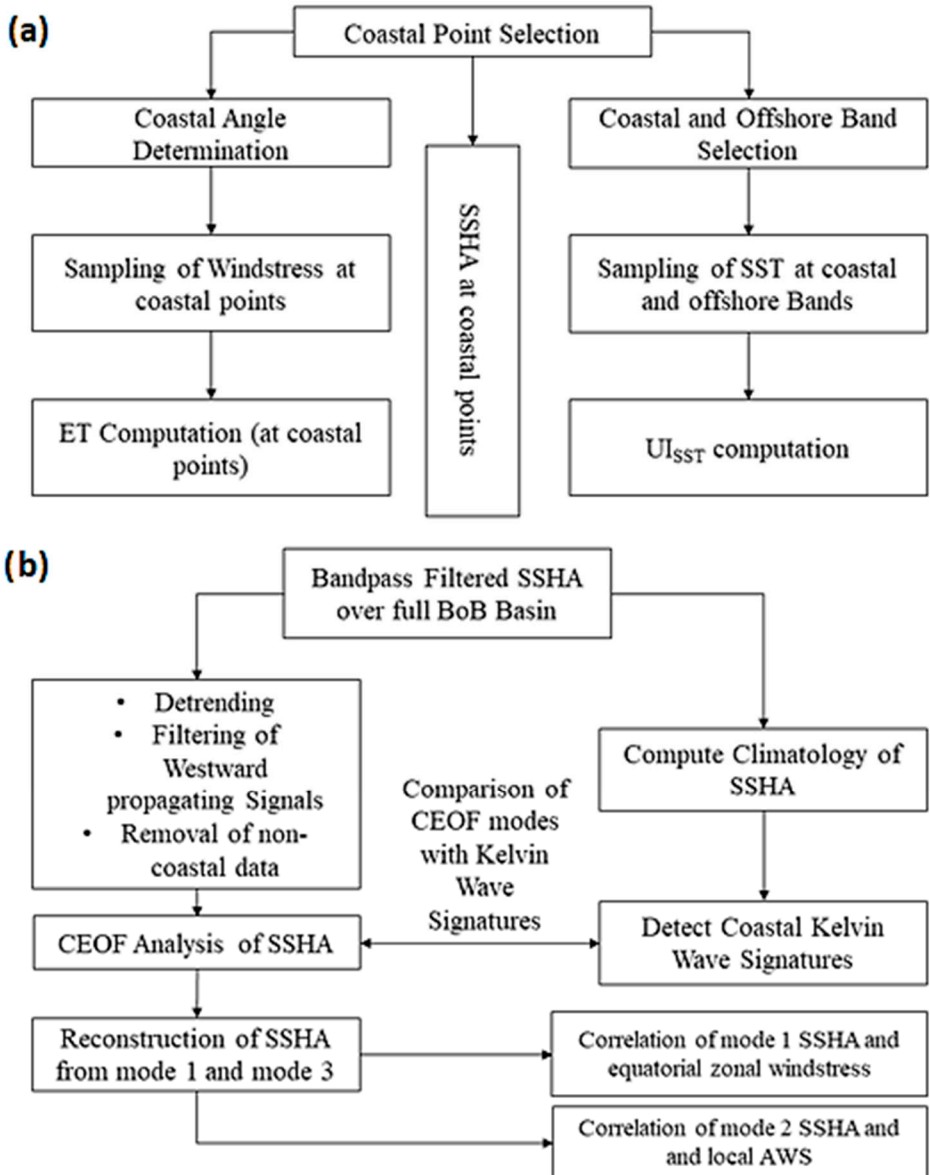

**Figure 2.** (**a**) Methodology of computation of upwelling indicators at the coastal points; (**b**) Analysis of spatial SSHA for detection of coastal Kelvin waves.

### 3.1. Estimation of Coastal Angles

The traditional method of the estimation of coastal angles involves a visual estimation of the slope of the coastline, as demonstrated by [21] for the west coast of North America over a 200 mile (322 km) section based on bathymetric charts. Ref. [22] used a similar method in the Canary current upwelling system utilizing the gridded relief model ETOPO1 based on the mean slope of the 200 m isobaths line over a segment of the length of roughly 80 km. Ref. [23] used a straight line inclined at 24° to approximate the western coast of India from 10° to 20°N. Ref. [24] extended the same approximation down to the southern tip of India in their analyses involving the use of coastal angles. Similar simplified approximations for the coastline were also used by ref. [25] in their analysis of coastal upwelling along the KwaZulu Natal Bight in South Africa. To determine the orientation of the coastline in the present analysis, at each of the 23 coastal points, the coast was partitioned into two lines that converged at that given point. The two lines on either side of the given point were then simplified using the Douglas–Peucker line simplification algorithm (with varying tolerance parameter values) [26] and the point closest to the given location was connected

to form a line segment parallel to the coast at that point. The orientation of the new line then determined the coastal angle. The coastal angle values for the different tolerance parameters are tabulated as Table 2. It may be noted that the tolerance value of 2.5° (in terms of the distance centered about a coastal point) resulted in the entire stretch of the coastline to be approximated as a single straight line, the simplest possible approximation. Three different approximations of the coastline around point 11 based on three different tolerance values are demonstrated in Figure 3. The ET corresponding to the lowest tolerance parameter value of 0.25° (5th column of Table 2) was used for the entire analysis. However, for comparative purposes, we also presented results for two tolerance values for coastline simplification, 0.25° and 1° (5th and 7th columns of Table 2), centered about the coastal points.

**Table 2.** Values of Coastal Angles corresponding to different tolerances at each coastal point.

| Point Identifier | Geographic Reference | Latitude | Longitude | Tolerance Value for Coastal Angle Determination (Distances) | | | |
|---|---|---|---|---|---|---|---|
| | | | | 0.25° | 0.5° | 1.0° | 2.5° |
| 1 | Kashinagara | 18.790 | 84.558 | 307.881 | 305.986 | 305.986 | 328.321 |
| 2 | - | 18.554 | 84.359 | 307.881 | 305.986 | 305.986 | 328.321 |
| 3 | Kalingapatnam | 18.309 | 84.129 | 307.881 | 305.986 | 305.986 | 328.321 |
| 4 | - | 18.026 | 83.557 | 307.881 | 305.986 | 305.986 | 328.321 |
| 5 | - | 17.590 | 83.215 | 307.881 | 305.986 | 305.986 | 328.321 |
| 6 | Tandava river mouth | 17.276 | 82.589 | 314.701 | 305.986 | 305.986 | 328.321 |
| 7 | Kakinada | 16.909 | 82.371 | 310.404 | 305.986 | 305.986 | 328.321 |
| 8 | Lanka Islands | 16.558 | 82.303 | 303.787 | 305.986 | 305.986 | 328.321 |
| 9 | - | 16.372 | 81.553 | 301.885 | 305.986 | 328.321 | 328.321 |
| 10 | - | 16.293 | 81.267 | 301.885 | 309.486 | 328.321 | 328.321 |
| 11 | - | 15.972 | 81.152 | 293.506 | 309.486 | 328.321 | 328.321 |
| 12 | - | 15.889 | 80.677 | 305.507 | 309.486 | 328.321 | 328.321 |
| 13 | Chinnaganjam | 15.672 | 80.264 | 329.142 | 331.124 | 328.321 | 328.321 |
| 14 | Kavali | 15.074 | 80.048 | 361.978 | 331.124 | 328.321 | 328.321 |
| 15 | - | 14.577 | 80.196 | 355.912 | 355.912 | 328.321 | 328.321 |
| 16 | - | 14.066 | 80.126 | 361.978 | 355.912 | 328.321 | 328.321 |
| 17 | Sriharikota | 13.782 | 80.257 | 352.805 | 355.912 | 349.842 | 328.321 |
| 18 | Ennore Port | 13.283 | 80.346 | 352.805 | 355.912 | 349.842 | 328.321 |
| 19 | Mamallapuram | 12.462 | 80.156 | 352.805 | 355.912 | 349.842 | 328.321 |
| 20 | Puducherry | 12.040 | 79.872 | 351.102 | 355.912 | 349.842 | 328.321 |
| 21 | - | 11.672 | 79.759 | 351.102 | 355.912 | 349.842 | 328.321 |
| 22 | Sirkazhi | 11.196 | 79.857 | 351.102 | 349.842 | 349.842 | 328.321 |
| 23 | Point Calimere | 10.308 | 79.880 | 341.777 | 349.842 | 349.842 | 328.321 |

### 3.2. Estimation of Upwelling Indices

The wind and SST parameters were sampled at the coastal points to first compute the ET and UI. The windstresses were sampled at each of the 23 coastal points using a Cressman window with a radius of influence of 75 km [27], and the ET was obtained from it, following [24], as:

$$\tau_a = -\frac{abs(lat)}{lat}\left(\tau_x\,cos\left(\theta - \frac{\pi}{2}\right) + \tau_y\,sin\left(\theta - \frac{\pi}{2}\right)\right) \tag{1}$$

$$\text{ET} = \frac{\tau_a}{\rho f} \tag{2}$$

where $\tau_x$ is the zonal component of wind stress, $\tau_y$ is the meridional component of windstress, $\theta$ is the coastal angle (angle subtended by the vector normal to the shoreline, pointing seaward, clockwise with reference to the east), $\rho$ is the density of water, $f$ is the Coriolis parameter, and ET is the Ekman Transport.

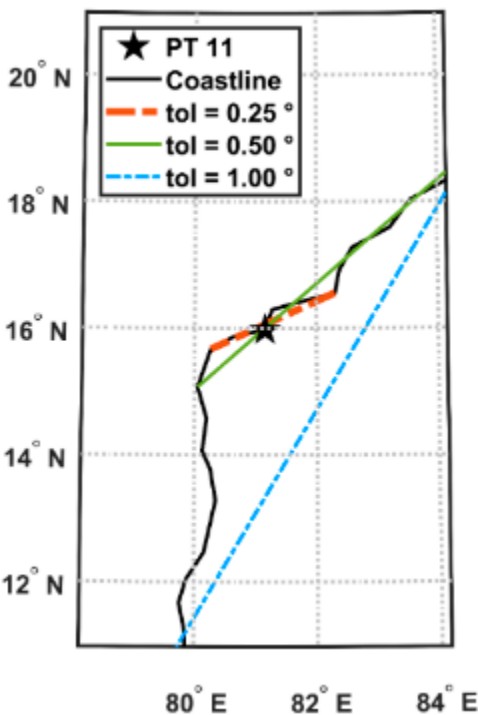

**Figure 3.** Linear approximations of the coastline at coastal point 11 for three different tolerance parameters of line simplification.

It was necessary to compute a nondynamical UI for intercomparison, and for this purpose we relied on a method similar to the traditional SST-based approach adopted by ref. [28] in their analysis of coastal upwelling along the SW coast of India [24]. The coastal upwelling was characterized based on the latitudinal temperature gradient given by the SST difference between a point on the coast and another point at a fixed distance offshore at the same latitude [10,28–32]. However, the limitations of using such an index based on a fixed longitudinal offset to analyze the BoB coastal upwelling system was well identified by ref. [11], which pertained to the considerable latitudinal variation of cross-shore width of the upwelling zone. Ref. [11] also observed the presence of cold waters offshore, which raised the possibility of underestimating the upwelling in the western BoB. To overcome this difficulty, they developed a more dynamic method of identifying the coastal and offshore bands based on the gradient of SST. In our analysis, the high-gradient region adjacent to the coast (with a gradient greater than $4 \times 10^{-4}$ °C/km) was identified as the coastal band at each latitude from the SST gradient climatology of July, following ref. [11]. The offshore band was identified as a 100 km-wide low-SST-gradient (uniform SST) region which is as close to the coast as possible.

The difference between the SST sampled at the coastal band and the SST sampled at the offshore band was the nondynamic $UI_{SST}$ [11], represented as:

$$UI_{SST} = SST_{coast} - SST_{offshore} \tag{3}$$

Each of the datasets (ET, $UI_{SST}$, and SSHA) were presented and compared in terms of a complete 10-year filtered time series or the decadal climatology. For the filtering of each time series data, a zero phase-shift third order bandpass Butterworth filter [33] was used with lower and upper thresholds of 0.90 cycles/year and 2.5 cycles/year, respectively. Strong peaks were observed around 1 cycle/year and 2 cycles/year in all the parameters in the wavelet spectra of the $UI_{SST}$, ET, and SSHA (Figures S1–S3 of Supplementary Materials).

*3.3. CEOF Analysis*

The EOF analysis is a particularly effective tool for analyzing strongly correlated fields (such as met–ocean observations) with the capability of reducing a data vector $x$ (consisting of $x_k$; $k = 1, \ldots, K$) into a set of M PCs denoted by $u$ (consisting of $u_m$; $m = 1, \ldots, M$), such that each element of $u$ is a linear combination of the elements of the anomaly of $x$ ($x' = x - \bar{x}$). These PCs satisfy two conditions: first, each PC captures the largest amount of variance of the data ($x$) (i.e., the first PC ($u_1$) explains the largest amount of variance, the second PC ($u_2$) explains the largest variance remaining after removing $u_1$, and so on), and second, they are uncorrelated with each other. The PCs are uniquely determined by the projection of the data anomaly vector ($x'$) on the eigenvectors of the covariance matrix of $x$ ($e_m$; $m = 1, \ldots, M$) [34].

$$u_m = e_m^T x' = \sum_{k=1}^{K} e_{k,m} x'_k; m = 1, \ldots, M \tag{4}$$

For the 2-dimensional geophysical data, the EOF analysis allows us to identify dominant modes of variability in a field which are coherent over large spatiotemporal scales. An EOF analysis could be used to decompose a field varying in both space and time into a set of fixed spatial patterns that vary in strength coherently over time. These fixed spatial patterns are the EOFs or PCs, while the temporal variation of each PC is given by its respective time series [35,36]. It is pertinent to mention that the EOF analysis, which effectively captures coherent variations, is particularly suited for studies involving planetary scale phenomenon and were extensively used to observe and analyze atmospheric and oceanic planetary scale activity [37–40]. Ref. [41] used an EOF analysis of SSHA to characterize equatorial intraseasonal Kelvin waves, while ref. [42] used an EOF analysis of the SSHA to characterize equatorial Rossby waves.

However, a key limitation of the traditional EOF analysis is its inability to identify phenomena involving signals with a spatiotemporal distribution which propagates spatially. To overcome this limitation, the Complex EOF (CEOF) analysis approach was adopted [43,44]. This approach starts with a Hilbert transform of the original data to generate a complex dataset: the real part of this complex data corresponds to the original data, while the imaginary part of it is a phase-shifted (90°) version of the original data. The spatial distribution of the phase of the CEOF denotes a temporal/phase lag in excitations at the different locations, while its magnitude denotes the amplitude of the excitations [45,46].

In our analysis, the SSHA spatial data was first filtered zonally to exclude westward propagating wavenumbers (associated with radiated and interior Rossby waves). Subsequently, it was filtered with a zero phase-shift third order bandpass Butterworth filter [33] with lower and upper thresholds of 0.90 cycles/year (1.1 year cycle) and 2.5 cycles/year (0.4 year cycle). The data filtering followed an approach closely resembling those adopted by refs. [41,42]. A CEOF analysis of the SSHA was previously used to observe Rossby waves radiated from coastally trapped Kelvin waves in the eastern Arabian Sea by [47]. We then carried out a CEOF analysis of the filtered SSHA to identify significant propagating modes of variability associated with coastally trapped Kelvin waves.

## 4. Results

*4.1. Spatiotemporal Variability of Coastal Upwelling System in the Western Bay of Bengal*

The temporal variation of the bandpass-filtered ET along the coast and offshore transport resulting from the AWS is presented in Figure 4 for two different values of the tolerance parameter used to compute the coastal angles. Figure 4a corresponds to a tolerance of 0.25°, while Figure 4b corresponds to a value of 1.0°. While a greater alongshore variation of ET was observed in the first case, both figures illustrate the ET to turn positive starting from March and persisting up to September along the major part of the Indian east coast. The strongest ET was observed between points 6 and 8 (17.276°N–16.558°N, i.e., around Kakinada, from the mouth of the Tandava river to the Lanka Island in the

northern Godavari Delta in Andhra Pradesh), often extending further northward to point 1 (Palasa-Kasibugga). For a lower value of the tolerance parameter (Figure 4a), the ET along a southern stretch of the coast (point 13: Chinnaganjam (15.672°N) to 17 (13.782°N close to Sriharikota off Lake Pulicat) showed a more rapid increase in April, often reaching its maximum value by May (premonsoon) and declining through the SW monsoon months. The ET computed from the low tolerance parameter coastal angles agreed more closely with the temporal variability of the $UI_{SST}$ (particularly around point 15, where the coastline has a convex curvature), as in Figure 5. The duration of peak $UI_{SST}$ signatures (around point 15) observed in our analysis also agreed closely with the findings of [11], which introduced the methodology for the $UI_{SST}$ computation utilized in our analysis. Hence, it was observed (by comparing Figure 4a,b) that an oversimplified coastline could lead to significant differences in the observed ET variability. We relied upon ET corresponding to the lowest tolerance parameter value of 0.25° for the analysis, as this also matched the grid spacing of the windstress data.

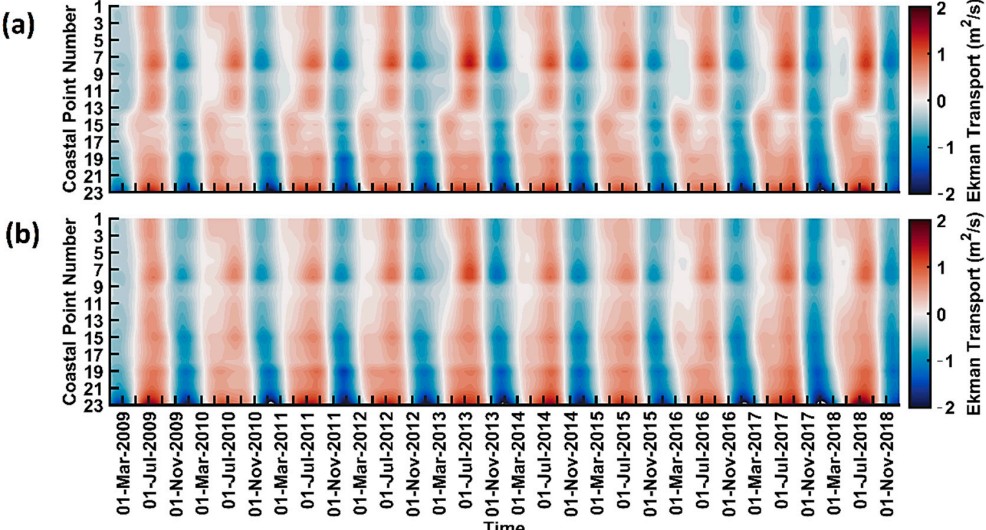

**Figure 4.** Temporal variation of ET along the coastline for coastal angles computed with tolerance parameter values of (**a**) 0.25° and (**b**) 1.0°.

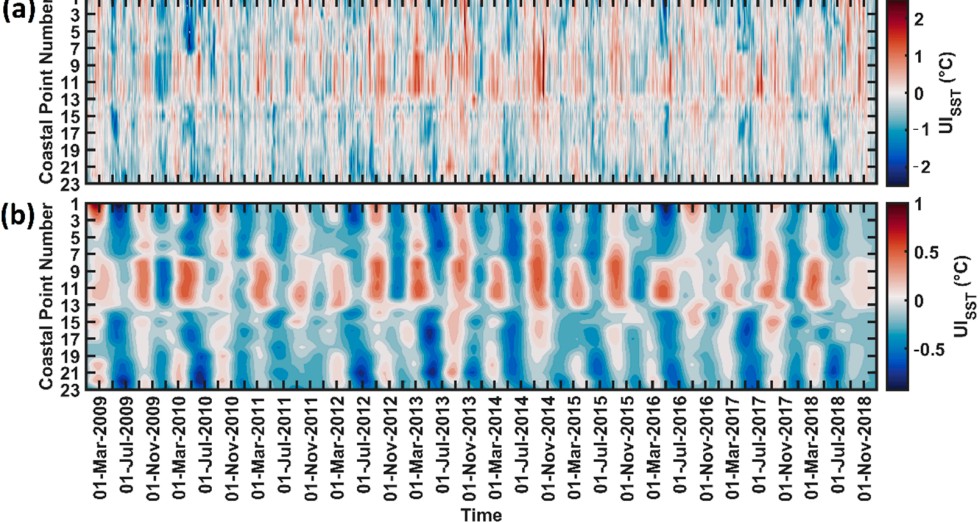

**Figure 5.** Temporal variation of (**a**) unfiltered and (**b**) filtered SST-based UI ($UI_{SST}$) along the eastern coastline of India.

The temporal variation of the filtered and unfiltered $UI_{SST}$ along the coast is illustrated in Figure 5. Most of the large-scale variability in the unfiltered data is captured by the bandpass-filtered dataset. In this analysis, a negative value of the $UI_{SST}$ indicates the presence of a relatively colder SST near the coast, a signature of coastal upwelling. Strong (negative) $UI_{SST}$ values were observed along the southern part of the coast from points 14 (15.074°N) to 23 (10.308°N) (Kavali to Point Calimere), and from point 1 (18.790°N) to 7 (16.909°N), i.e., between Kashinagara and Kakinada. The $UI_{SST}$ value peaked around May in the northern part of the coast (particularly around Kalingapatnam), and around Sriharikota (point 17) in the southern part of the coast. The $UI_{SST}$ peaked around June further south (south of the Ennore port (Point 18 in Figure 1b)). Coastal upwelling ceased along the northern part of the coast by July but extended up to August in the south (where the peak was also delayed). Though prominent upwelling signals (negative $UI_{SST}$) were not observed between points 9 (southern Godavari Delta) and 13 (Chinnaganjam), weak downwelling signals (positive $UI_{SST}$) persisted around point 12 throughout the SW monsoon period, with peak downwelling observed in March and October.

A decadal climatology (averaged over the years 2009 to 2018) of the filtered ET, $UI_{SST}$, and SSHA in the study region was generated (Figure 6). In the figure, the contours of the $UI_{SST}$ and SSHA were overlaid on the ET sampled at each coastal point. Coincident positive ET and negative $UI_{SST}$ (consistent with coastal upwelling) were observed along the southern part of the coastline (points 14–23, i.e., between Kavali and Point Calimere), and in the north, between point 1 and point 7 (from Kashinagara to Kakinada). However, a much closer overlap between $UI_{SST}$ and ET in the southern part of the coastline was observed compared to the northern one along the western BoB. The ET rose gradually from late April along the northernmost part of the coast (point 1 to 3) with a coincident decrease in the $UI_{SST}$. To the south of this, between point 4 and 7, the ET similarly increased from May, with a coincident drop in the $UI_{SST}$. Peak $UI_{SST}$ values were observed along the entire northern section of the coast (point 1–7) in June. Towards the end of June, as the $UI_{SST}$ values representative of upwelling begin to weaken, the ET continued to rise through July and August, particularly around Kakinada. In the southern part of the coast, both the $UI_{SST}$ associated with upwelling and the ET rose from March, reaching peak values in April between points 15 and 16 and points 22 and 23 in May. Between points 15 and 16, the ET decreased through June and July, with a coincident decrease in the $UI_{SST}$. Along the southernmost part of the coast (south of point 19 near Mamallapuram), and particularly to the south of point 21, strong ET values persisted from the beginning of May until the end of August. Strong negative $UI_{SST}$ were also observed in this region up to June, with a gradual decline through July and August. Thus, it was observed that the $UI_{SST}$ values representative of upwelling peaked much ahead of the peak ET values in the northern part of the coast (north of point 7). In the southern part, the peak of the $UI_{SST}$ occurred within an extended period of high ET. The southern stretch of the coast (points 19 (Mamallapuram)–22 (Sirkazhi)) showed coincident positive ET, weakly negative $UI_{SST}$, and negative SSHA in the later part of the SW monsoon season, while the northernmost part of the coast (from point 1–3) illustrated similar coincident SSHA values in the premonsoon period.

The sea levels and circulation in the BoB are determined by both local and remote wind forcing during the SW monsoon [14]. The prominent remote forcing mechanism in this region consists of two upwelling and two downwelling Kelvin waves which propagate counterclockwise along the coastal waveguide of the BoB [48]. These waves are generated near the equator due to the blocking of eastward-propagating equatorial Kelvin waves. In addition, coastal Kelvin waves could also be triggered by coastal upwelling along the margins of the Bay [49]. We therefore analyzed the coastally trapped wave activity along the western margin of the BoB based on the decadal climatology of seasonally filtered SSHA.

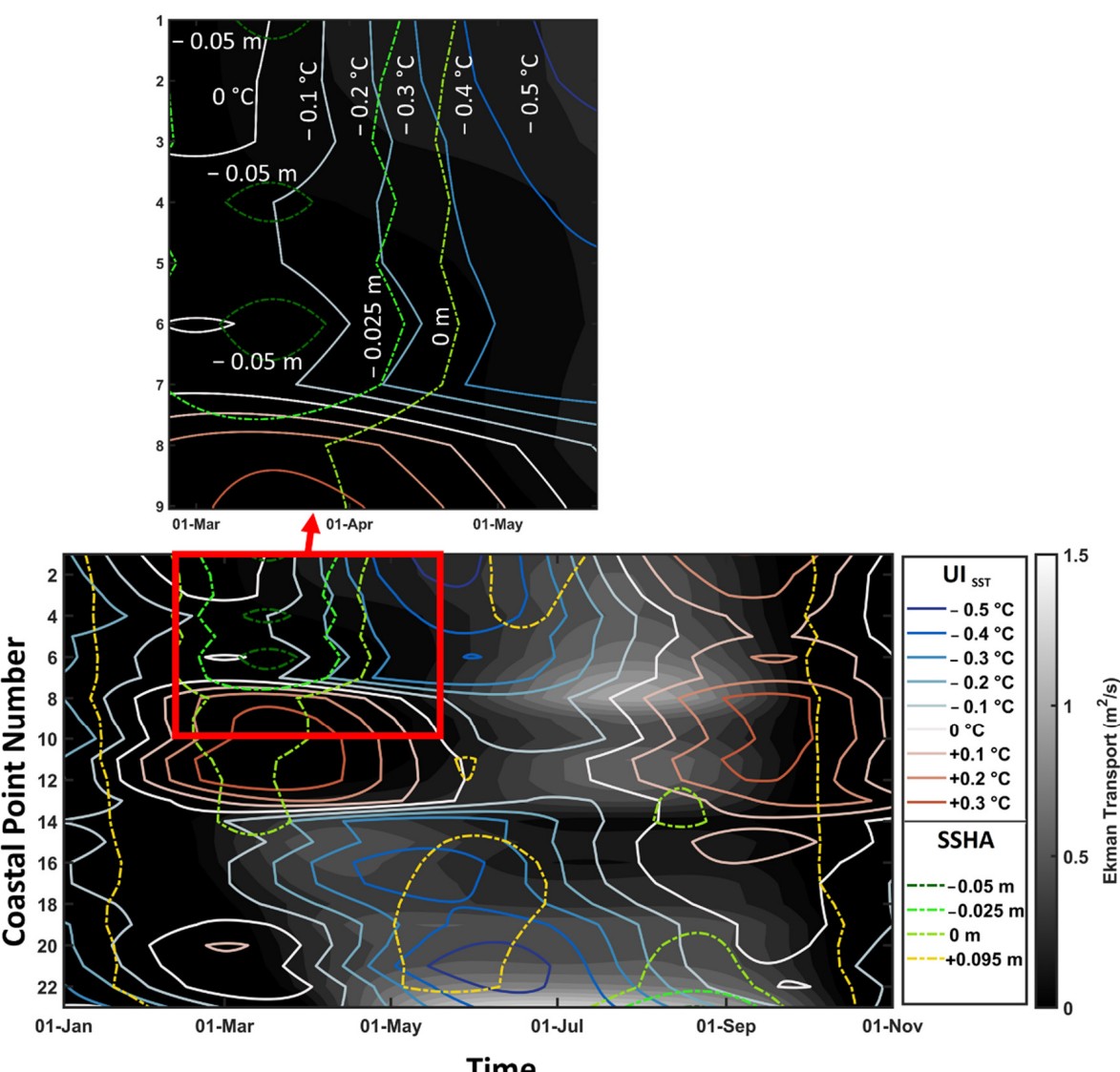

**Figure 6.** Climatology of ET with UI$_{SST}$ contour (solid lines) and SSHA contours (dashed line) considering all the bandpass-filtered variables. Inset: Premonsoon UI$_{SST}$ and SSHA in the northern section of the coast.

### 4.2. Coastally Trapped Kelvin Wave Activity

SSHA variations contain distinct signatures of propagating planetary-scale waves including the subsurface variations associated with them [50]. Following [13], we first located the activity of upwelling and downwelling coastally trapped Kelvin waves in the region using the long-term mean SSHA variability. Upwelling Kelvin waves are associated with negative coastal SSHA, while downwelling Kelvin waves are associated with positive coastal SSHA signatures. The 10-year mean (climatology) SSHA from 2009 to 2018 are presented in Figures 7 and 8 for every 10th day starting from the initiation of the wave. It was observed that the coastal BoB experienced two sets of upwelling and downwelling Kelvin waves that propagated counterclockwise starting from the coast of Sumatra in the east up to the coast of India. The first upwelling (downwelling) Kelvin wave prevailed from January to April (end of April to late August), and the second upwelling (downwelling) Kelvin wave from the end of August to September (October to December). The pre-SW monsoon period (months of March, April, and May) and SW monsoon period (the months of June, July, August, and September) considered covered the period of activity of three of the Kelvin waves: the first upwelling and downwelling and the second upwelling Kelvin

wave [13]. Since the second downwelling Kelvin wave was restricted to the southeastern BoB and terminated before reaching the head Bay, it was unlikely to affect the coastal upwelling along the western BoB.

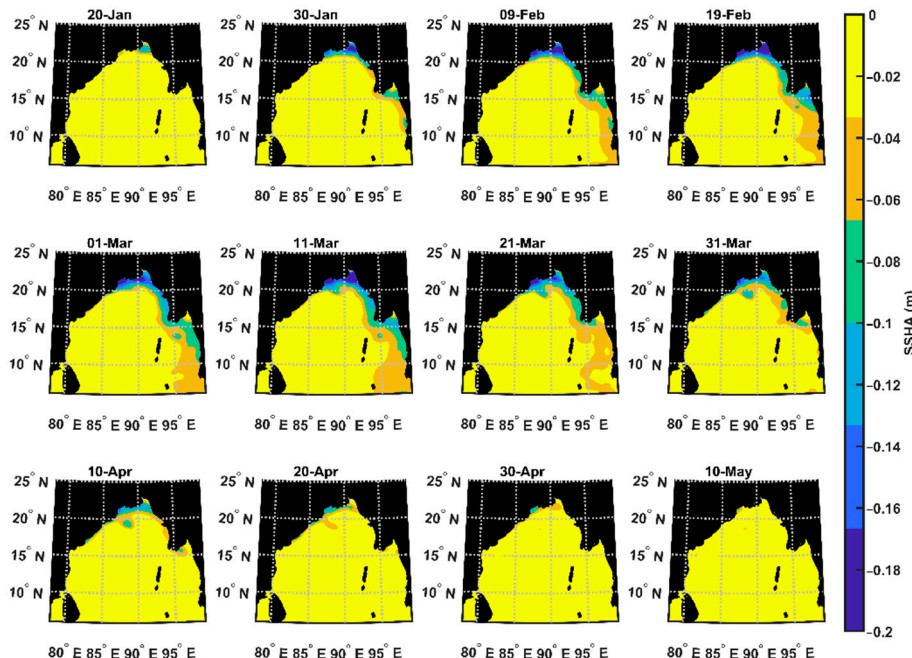

**Figure 7.** Decadal Climatology (2009 to 2018) of positive SSHA corresponding to the 1st upwelling Kelvin wave activity.

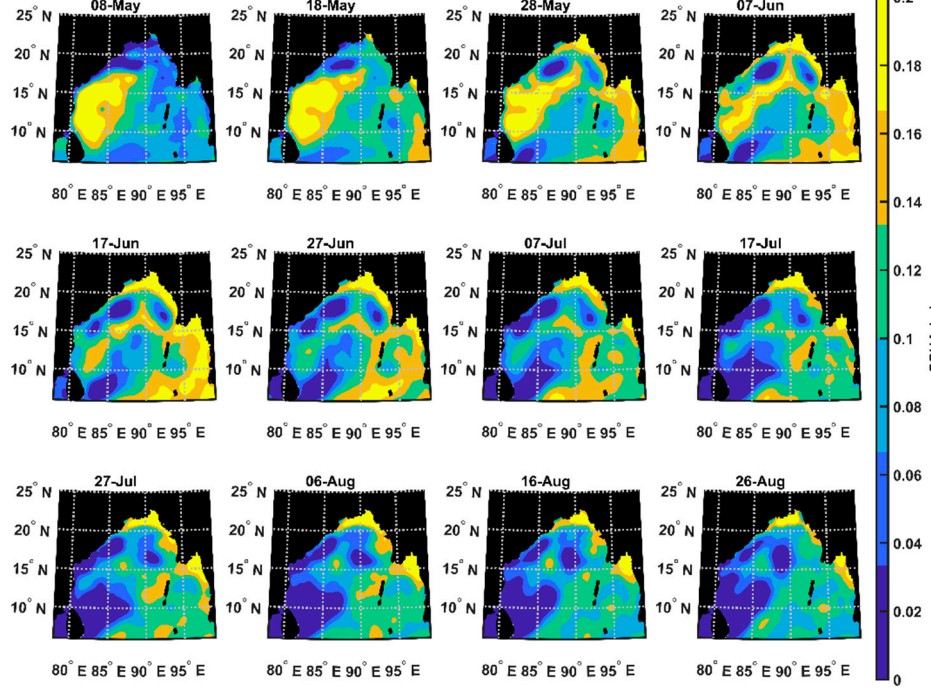

**Figure 8.** Decadal Climatology (2009 to 2018) of negative SSHA corresponding to the 1st downwelling Kelvin wave activity.

Negative SSHA associated with the first upwelling Kelvin wave appeared along the head and eastern margins of the BoB by the end of January. The southward extent of these signatures along the western margin of the Bay was restricted to the 20°N latitude until

the earlier part of February. These signatures continued to intensify and a narrow band of weak upwelling signatures (approximately −0.005 m SSHA) extended down to Kakinada (point 7) by the first week of March. By the end of March, stronger upwelling signatures (<−0.01 m SSHA) extended down to Kakinada, with weaker upwelling signatures (negative SSHA) observable down to point 10. These SSHA signatures started to weaken in April and the weakening signatures receded back to the 20°N latitude at the head of the Bay by the beginning of May (Figure 7). Positive SSHAs were observed prominently from mid to late May at the head Bay region along the coast of West Bengal and Bangladesh and along the eastern BoB. These anomalies could be observed in the region throughout the season, persisting up to the middle of August. Within the first half of June, a narrow continuous band of downwelling SSHA signatures (approximately 0.01 m) extended down to 17.5°N. These coastal signatures began to weaken in the later half of June and receded back to the 20°N latitude by early July (Figure 8.) Negative SSHA signals first appeared along the western boundary of the BoB, along the coast of India, near 15°N in early August. The anomalies then propagated southward, first along the coast of India and then around Sri Lanka, into the Arabian Sea. Since then, it was difficult to isolate the signal corresponding to coastally trapped Kelvin wave activity from the SSHA fields directly, and a CEOF analysis of the SSHA data was carried out to observe these variations distinctly. In the following sections, we attempt to identify the PCs of the SSHA associated with these Kelvin wave signatures.

### 4.3. CEOF Analysis of SSHA

The first CEOF mode (Figure 9), which captured 75.9% of the variation in the coastal SSHA, consisted of a band of strong magnitudes spread along the coastal waveguide of the BoB with a progressive narrowing of the band observed counterclockwise along the waveguide. The phase of the first CEOF also exhibited a small clockwise rotation from the eastern to the western margin of the bay. The greatest magnitude (>0.035) of this CEOF was observed at the head bay with weaker values (around 0.03) along the eastern and northwestern margin. Along the east coast of India, these signals extend down to Kakinada (point 7). The magnitude and phase of the corresponding time series illustrated a roughly annual cycle, with magnitude peaks (crests) observed from February to March (August–October). A clockwise rotation (phase decrease) with time (covering roughly 360° per year) was also observed. Two patches of high offshore CEOF values were observed in Figure 9a; however, these were spatially disconnected from the coast, illustrating large discontinuous phase differences. The decadal climatology of the SSHA reconstructed from the first CEOF mode at coastal points 2, 7, 12, 17, and 22 are presented in Figure 9d. A double-peaked variation was observed, with the larger peak (crest) in May–June (March–April), and a smaller peak (crest) in October–November (August–September). The magnitude of the variation was observed to decrease southward along with an increasing time lag.

The second CEOF mode (Figure 10), which captured 15.7% of the variation in the coastal SSHA, consisted of coastally bound signatures along the western margin of the BoB (extending from the southern coastline of Sri Lanka northward up to 18°N) with a slight clockwise rotation from the north to the south. The magnitude of the principal component illustrated an annual cycle with a peak in late August and its phase rotated 720° in a year with a faster 360° rotation from December to May, and a slower 360° rotation from May to December. The climatology of the reconstructed SSHA showed a double-peaked structure similar to the first mode. A larger peak (crest) is observed in November–December (July–August), while a smaller peak (crest) appears in April–May (February–March). This climatology illustrated the weakening of signals towards the north along with a phase lead. The larger crest appears at point 22 (Sirkazhi, 11.196°N) on 28th July with a magnitude of −0.05, while the corresponding crest for point 2 (18.554°N) appears on 12th August with a magnitude of −0.02.

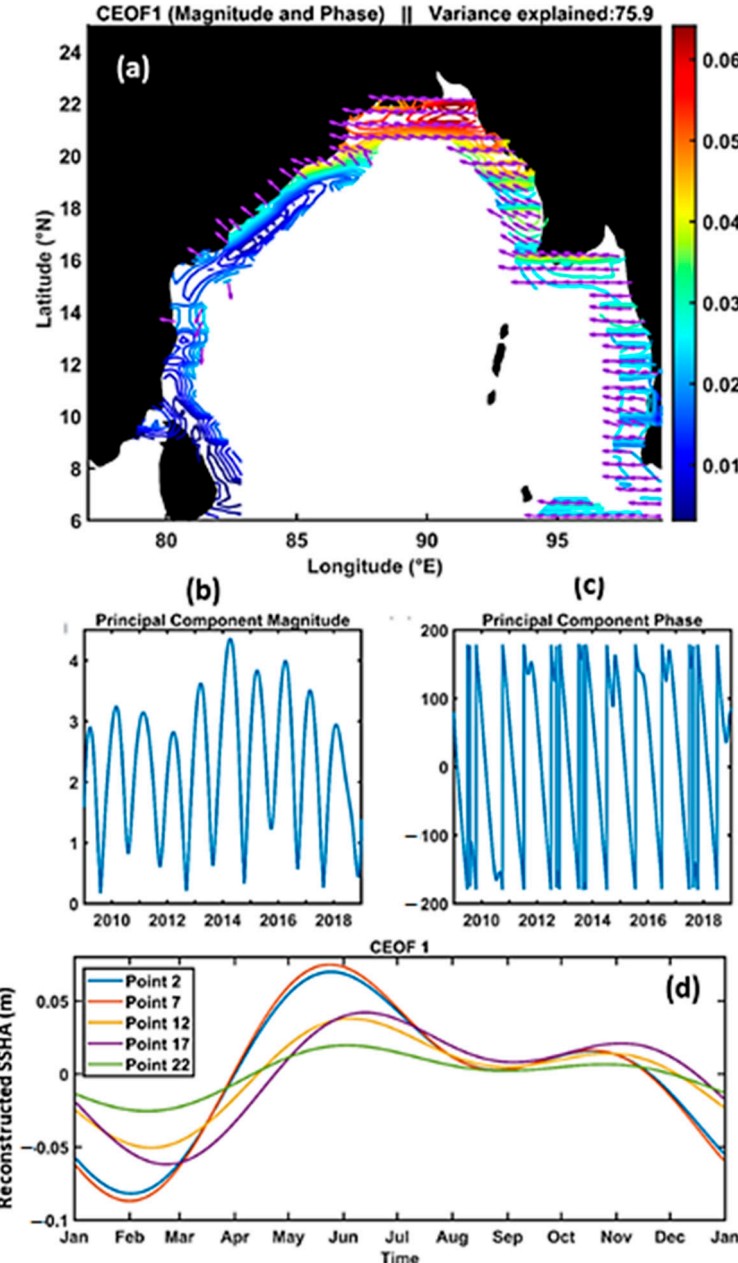

**Figure 9.** First mode of SSHA in the study region: (**a**) Magnitude (colored contours) and phase (purple arrows) of EOF; (**b**) Magnitude of PC, (**c**) phase of PC, and (**d**) climatology of the reconstructed coastal SSHA around points 2, 7, 12, 17, and 22.

Refs. [13,44] observed that equatorial zonal wind forcing triggers two pairs of up-welling and downwelling equatorially trapped Kelvin waves along the coastal waveguide of the BoB. Hence, to ascertain the driver of this mode, the correlation coefficient between the equatorial zonal wind forcing and the SSHA at each coastal point reconstructed from the first CEOF mode was computed (Figure 11). The greatest positive correlation (>0.86) is observed between point 1 (Kashinagara, 18.790°N) and 10 (16.293°N), with the greatest value (0.87) observed at point 5 (17.950°N, north of the Tandava river). Similarly, as the second CEOF closely resembled the coastal SSHA response associated with coastal up-welling, the correlation coefficient between the local AWS and the SSHA at each coastal point reconstructed from the second CEOF mode was also computed (Figure 12). As a positive AWS results in coastal upwelling, which is associated with a negative SSHA [1,2], this correlation is expected to be negative if the second CEOF mode is indeed associated

with coastal upwelling. Figure 12 illustrates the strong negative correlation (i.e., less than −0.70) between point 5 (17.590°N, north of the Tandava river) and 12 (15.889°N, north of Chinnaganjam), with the strongest correlation of −0.80 at point 11 (15.972°N). A slightly weaker negative correlation (<−0.65) was also observed at points 2–4 (18.554°N–18.026°N) and 13 (Chinnaganjam, 15.672°N).

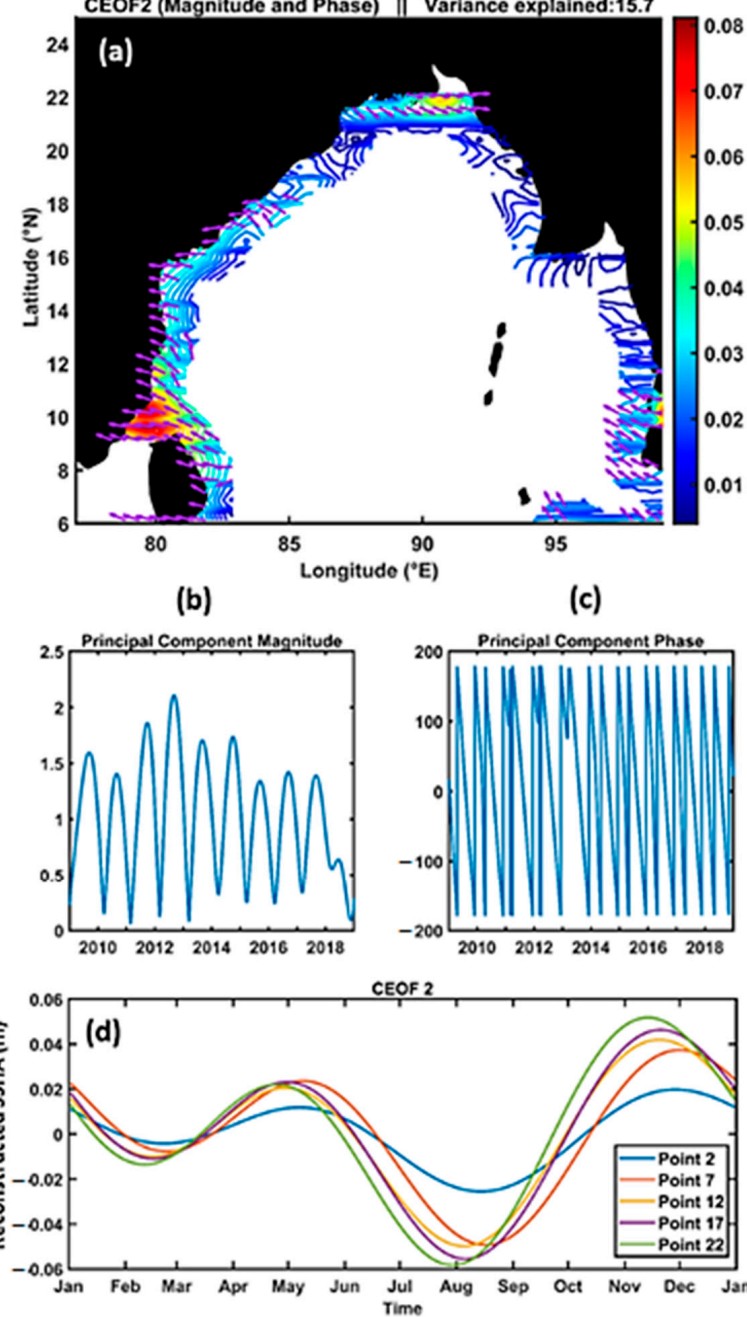

**Figure 10.** Second mode of SSHA in the study region: (**a**) Magnitude (colored contours) and phase (purple arrows) of EOF; (**b**) Magnitude of PC, (**c**) phase of PC, and (**d**) climatology of the reconstructed coastal SSHA around points 2, 7, 12, 17, and 22.

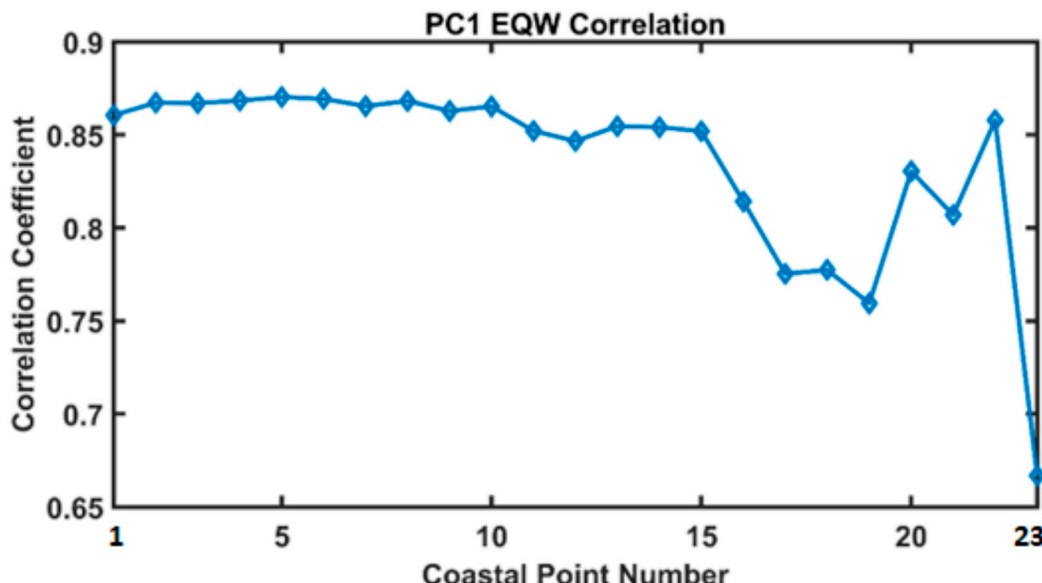

**Figure 11.** Correlation between the SSHA reconstructed with the first mode and equatorial zonal wind forcing.

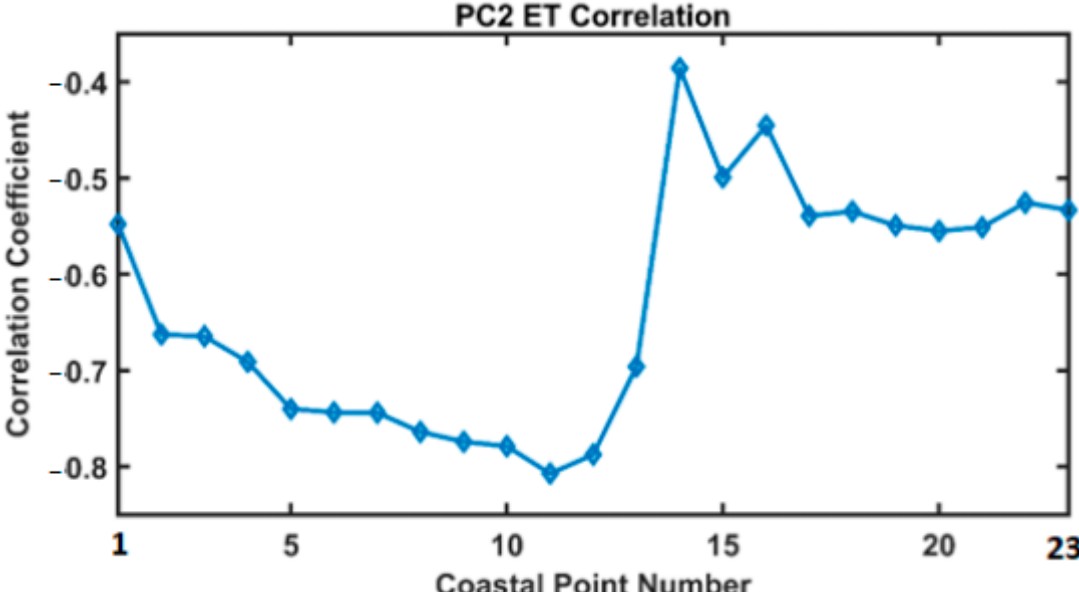

**Figure 12.** Correlation between the SSHA reconstructed with the first mode and local ET.

## 5. Discussions

In our analysis of the western BoB coastal upwelling system, the temporal variation of the ET (Figure 4) and the $UI_{SST}$ (Figure 5) were first analyzed. While comparing the climatological distribution of the ET and $UI_{SST}$, collocated instances of high-positive ET and negative UI (both indicative of coastal upwelling) were observed at different stretches of the coast during the SW monsoon season, with two main peculiarities evident in the analysis (Figure 6). First, the northern part of the coast (between point 1 and point 7 from Kashinagara to Kakinada) illustrated a poorer agreement between the $UI_{SST}$ and ET. This anomaly was observed most clearly in July (Figure 6), when the ET in the north illustrated an increasing trend while the corresponding $UI_{SST}$ signals illustrated a diminishing trend. Refs. [51–53] attributed the inhibition of coastal upwelling to freshwater influx from river discharge, which exacerbated stratification in the northern part of BoB, primarily from

August. Hence, our observed weakening of UI$_{SST}$ signatures starting from late June and amplifying further in July cannot be associated with freshwater influx (Figure 6). The second peculiarity observed in the climatology (Figure 6) was the absence of a negative SSHA (typically associated with coastal upwelling systems) along the Indian east coast.

The SSHA variations during premonsoon and the SW monsoon months showed the existence of both the first upwelling (Figure 7) and first downwelling (Figure 8) Kelvin waves. The cross-shore width of these propagating Kelvin wave signals were much wider along the eastern margin of the BoB compared to the western margin (possibly due to the radiation of westward propagating Rossby waves). This also made the identification of the Kelvin wave propagation along the eastern coast of India difficult, and hence a magnified version of Figure 7 was presented as Figure 13 for better illustration. The peak upwelling Kelvin wave activity was observed along the western margin of the Bay in March (Figures 7 and 13). From the climatology presented in Figure 6, this was observed to be almost simultaneous with the appearance of a negative UI$_{SST}$ along the northern section of the coast. This preceded the appearance of a positive ET (particularly between point 4 and 7). The negative coastal SSHA associated with the first upwelling Kelvin wave was also prominently observed, with the SSHA contour of −0.025 m extending from point 1 down to point 7 in March (Figure 6). This indicated the triggering of coastal upwelling in the northern part of the coastline by the first upwelling Kelvin wave. The rising local AWS further strengthens and sustains coastal upwelling.

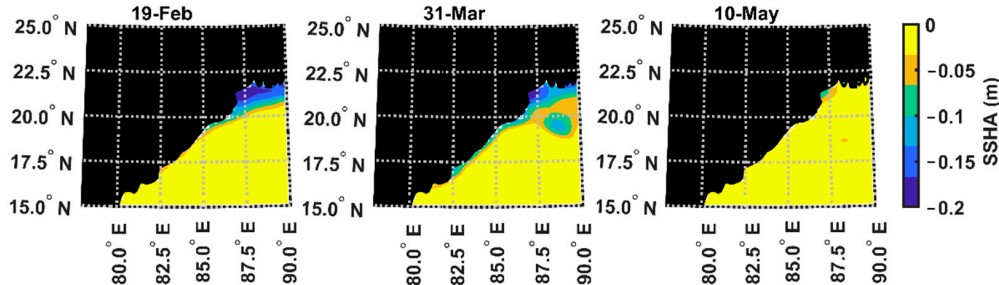

**Figure 13.** Decadal climatology (2009 to 2018) of unfiltered positive SSHA in the northwestern BoB during the first upwelling Kelvin wave period.

The propagation of the first downwelling Kelvin wave into the BoB was found to be less extensive than that of the first upwelling Kelvin wave (Figures 13 and 14). This intrusion to the south of point 4 (18.026°N) was evident in June on magnifying Figure 8 (presented as Figure 14). Positive SSHAs extended from point 1 to point 4 during June (Figure 6). This coincided with the contrast between the UI$_{SST}$ and ET, as previously discussed. The negative UI$_{SST}$ between point 1 and 4 continued to weaken while the local AWS (and ET) increased. To the south of the 18°N latitude line, from points 5 to 7, the onset of rising ET was slightly delayed. The UI$_{SST}$ observed here was weaker than that observed between points 1 and 4. A dark blue patch (representing a weakly negative SSHA) could be detected in the northwest BoB (near 17.5°N, 87.5°E) from early May to July. Figures 8 and 14 also demonstrated an offshore area of negative SSHA, possibly related to a cyclonic eddy; such an eddy could potentially modulate the southward propagation of the first downwelling Kelvin wave.

Figure 8 illustrated a large patch of positive SSHA in the southwest BoB that weakened progressively from May and ultimately dissipated within the first half of June. This might be associated with the anticyclonic gyre formed in the mid-February under the influence of the first upwelling Kelvin wave [44]. Ref. [44] also observed that this anticyclonic gyre reached maximum extent in early May and dissipated subsequently. The strong positive SSHA observed from May to June, between 15 and 22 in Figure 6, is related to this anticyclonic gyre. A patch of negative SSHA also developed in the southern BoB off the east coast of Sri Lanka (to the south of 10°N and west of 85°E) from early June. By August, it expanded

northward and eastward and reached the 15°N latitude along the east coast of India. This may be related to the pair of cyclonic eddies which form off the southeast coast of India around this time [44]. The associated negative SSHA can be noticed in Figure 6, to the south of point 20, in late August and early September. The SSHA variations associated with the coastally trapped Kelvin waves and those linked to coastal upwelling, and the interaction between the two cannot be discerned clearly through simple time series or climatological analyses. As the coastal upwelling signals were obscured in the north by coastally trapped Kelvin waves and in the south by the anticyclonic gyre, a CEOF analysis was employed to isolate SSHA variations associated with each phenomenon and analyze their interaction.

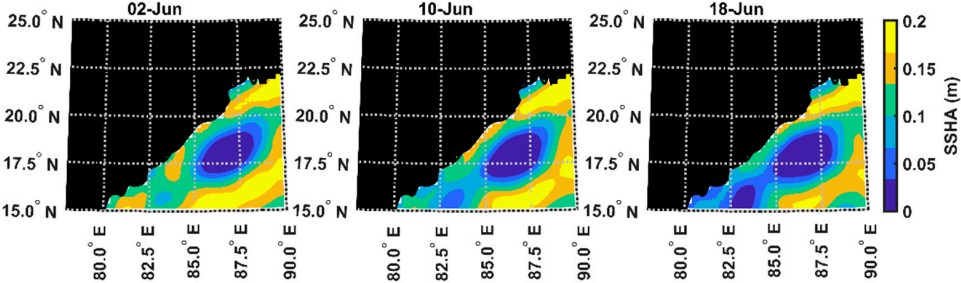

**Figure 14.** Decadal Climatology (2009 to 2018) of unfiltered negative SSHA in the northwestern BoB during the first downwelling Kelvin wave period.

The first mode of the CEOF consisted of coherent coastally trapped SSHA patterns propagating counterclockwise along the coastal waveguide of BoB, southward from the head of the Bay down to the 17°N latitude (Figures 9 and 15). Further, this mode was found to be associated with equatorial zonal wind forcing, which are known to drive coastally trapped Kelvin waves. The SSHA reconstructed from the first CEOF at coastal points 1–10 (18.790–16.293°N) were found to be correlated with equatorial zonal windstress, with a correlation coefficient greater than 0.86 (greatest value of 0.87 at 17.590°N). The larger peak (positive SSHA) and crest (negative SSHA) of this mode occurred during the first upwelling and downwelling Kelvin wave activity, respectively, as identified from the decadal climatology of SSHA. Thus, a close spatial and temporal agreement could be established between the two Kelvin waves and the first mode, along with a high correlation with its driver. The BoB is influenced by other large-scale propagating waves, such as the westward propagating Rossby waves radiated from coastally trapped Kelvin waves and from the interior Bay. However, as the westward-propagating signals were filtered prior to the CEOF analysis, the identified modes are unlikely to be related to westward-propagating Rossby waves. Moreover, no continuously propagating signals are incident on the east coast of India from the open ocean. This illustrated the SSHA variation related to Kelvin wave activity.

The second CEOF mode consisted of local ET (or AWS) driven SSHA closely resembling the expected SSHA response to ET with a weak southward propagation. This southward propagation was very likely associated with the lag in peak $UI_{SST}$ in the south (Figure 6). The climatology of the SSHA reconstructed from this mode turned negative during May–June, the period of peak upwelling in terms of the $UI_{SST}$. As the SSHA variations associated with this mode extend beyond 18°N, a strong possibility of interaction between the first and second CEOF modes arise between coastal points 1 and 7. We further considered this possibility by comparing individual contributions of the first mode, the combined contribution of the first two modes, and the total SSHA variability at point 5 (Figure 16), which is located centrally between point 1 and 7.

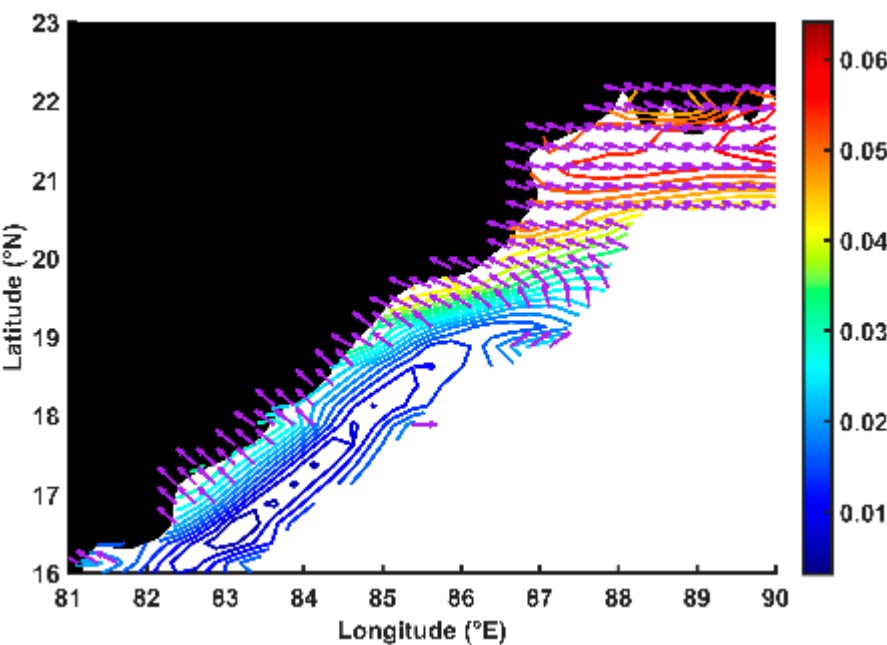

**Figure 15.** Magnitude (colored contours) and phase (purple arrows) of first mode of CEOF in the northwest BoB.

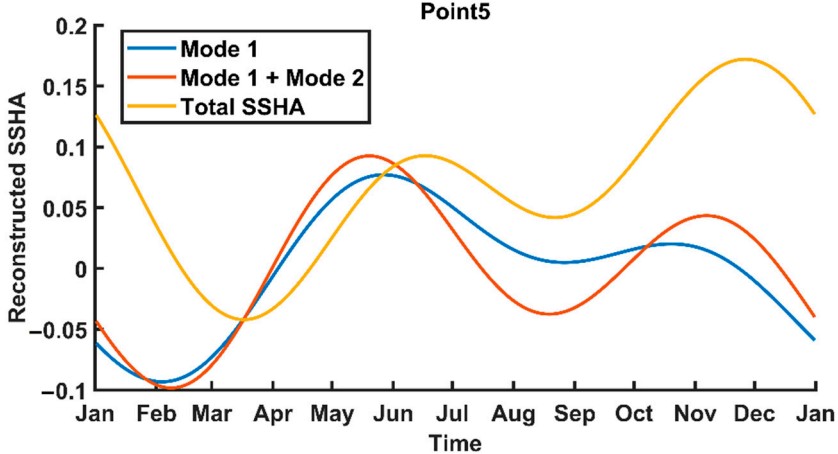

**Figure 16.** Climatology of reconstructed CEOF modes 1, the combination of mode 1 and 2, and the total SSHA.

Figure 16 illustrated that the combined SSHA and total SSHA are largely determined by the first mode throughout the premonsoon period (March, April, and May). The total SSHA closely follows the variation of the first mode with a lag of approximately a month. The negative peak of the first mode corresponding to the first upwelling Kelvin wave lowered SSHA in the region in March. As discussed previously, this coincided with period of strengthening negative $UI_{SST}$ signatures (in the absence of a strong positive ET). From the beginning of June, the contribution of the second mode (to the combined SSH) became progressively more important as the second mode approached its crest and the first mode waned from its peak value. The SSHA variability at point 5 from April to June (the period of rising $UI_{SST}$) were determined dominantly by the variability of the first mode. The first downwelling coastal Kelvin wave driven strengthening of the first mode produced a sharp increase in SSHA in May, the peak upwelling month. This resulted in a corresponding positive peak in SSHA the following month, which coincided with the period of anomalous cessation of coastal upwelling (as observed in the $UI_{SST}$ in the

presence of the strong, rapidly increasing ET). This may be associated with coastal Kelvin wave driven thermocline deepening, which restricts coastal upwelling [44].

## 6. Conclusions

The seasonal cycle of coastal upwelling along the northeast coast of India (between Kashinagara and Kakinada) during the premonsoon and the southwest monsoon seasons were analyzed on the basis of remotely sensed windstress, SST, and SSHA. The region displayed some anomalous patterns compared to the southeast coast (between Kavali and Point Calimere), which illustrated the characteristics of a typical local AWS-driven coastal upwelling system (Figure 6). Along the northeastern part of the coast, the onset of the SST signatures associated with coastal upwelling (negative UI$_{SST}$) were detected from April, though the local ET remained fairly low at the time. These SST signals continued to intensify and reached peak values by May, and subsequently declined through June and July. Surprisingly, this period of weakening upwelling-favorable SST signals coincided with a sharp rise in ET, the upwelling-favorable coastal wind forcing (Figure 6).

Coastal upwelling events are typically associated with negative coastal SSHA; however, the complex circulation of the BoB, characterized by numerous eddies and gyres, obscured these signals. In the southern part of the coast (between Kavali and Point Calimere), the negative SSHAs are suppressed by an anticyclonic gyre that formed in response to the first upwelling Kelvin wave and strengthened until early May. By contrast, the northern part of the coast illustrated strongly negative SSHAs prior to and during the period of initiation of coastal upwelling (March–April), which increases successively. This illustrated strong positive values by the time the UI$_{SST}$ signals started weakening (June). This anomalous behavior of the western BoB coastal upwelling system observed in the climatological distribution were investigated through a CEOF analysis of coastal SSHA.

The CEOF analysis revealed two modes of variation, which are associated with the remotely forced coastal Kelvin wave, and local ET-driven coastal upwelling, respectively. These two modes collectively explain most of the variability (91.6%) of the coastal SSHA along the margin of the BoB. The climatology of these two modes indicated that the strong negative SSHA observed around the time of initiation of coastal upwelling along the northeastern coast of India resulted from the propagation of the remotely driven first upwelling Kelvin wave, which possibly plays a role in triggering coastal upwelling along the northern part of the coast. The strong positive SSHA observed along the northeastern coast during the period of subsiding upwelling-favorable UI$_{SST}$ signals are found to be associated with the first downwelling Kelvin wave, which might be involved in its suppression. Coastally trapped upwelling (downwelling) Kelvin waves are known to trigger (suppress) coastal upwelling through modulations of the thermocline depth [44]. Thus, the seasonal coastal upwelling along the northeast coast of India was triggered and terminated by remotely forced coastal Kelvin waves. The local ET-driven second mode of coastal SSHA variability consisted of negative coastal SSHA coincident with the period of high local AWS. The existence of this mode established that a mode consisting of a negative SSHA was excited in response to local ET, while the coastal SSHA remained positive throughout. The second CEOF mode also illustrated strong negative values during June, and thereby illustrated that the upwelling along the southeastern coast is primarily local ET-driven.

**Supplementary Materials:** The following supporting information can be downloaded at: https://www.mdpi.com/article/10.3390/rs14194703/s1, Figure S1: The time-frequency wavelet spectra and time-averaged wavelet spectra of ET for five points along the east coast of India, Figure S2: The time-frequency wavelet spectra and time-averaged wavelet spectra of UI$_{SST}$ for five points along the east coast of India, Figure S3: The time-frequency wavelet spectra and time-averaged wavelet spectra of SSHA for five points along the east coast of India.

**Author Contributions:** S.R. and D.S. conceptualized the work; S.R. carried out the analysis. S.R. and D.S. wrote the manuscript with inputs from M.M.A. and M.A.B. All the authors contributed to revising the manuscript. All authors have read and agreed to the published version of the manuscript.

**Funding:** No external funding was received for carrying out the work.

**Data Availability Statement:** All the datasets used in the study are available free of cost through their respective websites.

**Acknowledgments:** The computation and data analysis has been carried out on Matlab 2021a with the Climate Data Toolbox [54] and the Douglas–Peucker line simplification algorithm implemented in Matlab by Schwanghart [55]. This study has been conducted using E.U. Copernicus Marine Service Information; https://doi.org/10.48670/moi-00185, https://doi.org/10.48670/moi-00168, and https://doi.org/10.48670/moi-00148. The authors thank IIT Bhubaneswar for facilitating the research work. SR was partially supported through a fellowship from SAC/ISRO and gratefully acknowledges the same. MAB's contribution was funded by NASA Physical Oceanography via the Jet Propulsion Laboratory (Contract #1419699). The authors are also thankful to the editorial board for the comments and suggestions, which contributed to the technical improvement of the manuscript.

**Conflicts of Interest:** The authors declare no conflict of interest.

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
