# Peer review of "Coastal Upwelling in the Western Bay of Bengal: Role of Local and Remote Windstress"

_remotesensing, doi:10.3390/rs14194703_

Round 1
Reviewer 1 Report (Previous Reviewer 1)
The authors have addressed most of my major concerns and provided detailed replies to my comments, which are to my satisfaction. I appreciate the implementation of new methods for SST-based index and CEOF analysis and the detailed description of the methods. I appreciate the introduction of recent results. The revised results section looks very reasonable to me. The revised manuscript deserves publication, but the readability still needs improvement before publication. Please see the attachments.

Author Response
Dear Reviewer,
The responses are uploaded in the document.
with regards
Corresponding Author

Reviewer 2 Report (Previous Reviewer 2)
Dear Authors
The manuscript is accepted in this present form.
Author Response
Reviewer Comments: Dear Authors, The manuscript is accepted in this present form.
Response: We deeply appreciate the time and effort spent in reviewing this manuscript and providing useful suggestions. We are glad to present an analysis that meets the expectations of the esteemed reviewer and the standards of the journal.
This manuscript is a resubmission of an earlier submission. The following is a list of the peer review reports and author responses from that submission.
Round 1
Reviewer 1 Report
The authors of the paper “Coastal Upwelling in the Bay of Bengal: Role of Local and Remote Windstress” have used a novel method for estimating the offshore Ekman transport and investigated the impact of local and remote wind stress on the upwelling strength over the western Bay of Bengal (BoB). While the topic is of interest to the ocean community, I believe it requires many substantial improvements before being published.
In particular, I found that the methods are not sound. This study also lacks a detailed background description of changes in upwelling and its associated mechanism over the BoB region. Some conclusion is not plausible. Some plots are hard to decipher, the language is not consistent/precise, and full of mistakes and typos.
I would suggest the authors should address the following issues for further improvement of this study. The present paper needs improvement of analysis and presentation of description in order to be deserved for publication. Therefore, it is my opinion that the manuscript should be rejected. I fairly encourage the author to improve the manuscript for resubmission.
Please see my detailed comments in the attachment.

Reviewer 2 Report
The manuscript, in general, is well written and provides some interesting
results. I have a few minor comments or concerns. The paper demonstrates a significant contribution with revision and clarification. The following are some minor comments that need to be addressed before the manuscript can be considered for publication.
The English should be revised carefully for clarity and to remove redundancy before a resubmission.
The endmembers of the Y-axis labels should be shown in Figures 4, 5, 6, and 12.
Figure 1(a): The label bar title needs to be clear. The label can be changed to km units.
Figure 4: The font size of figure captions (a) and (b) are inconsistent with other figures.
Figure 6: The figure is unclear. May reconsider the label color bar.
Figures 7 and 8: The X-axis labels are unclear.
Figure 9: The X-axis label is unclear. Also, the figure captions (a), (b), and (c) are placed inconsistently compared to previous figures.
Figure 10: The endmember of the X-axis label in figure 10 (b) is missing. The font size of the Y-axis label needs to be reconsidered.